# Disinfection Kinetics of Free Chlorine, Monochloramines and Chlorine Dioxide on Ammonia-Oxidizing Bacterium Inactivation in Drinking Water

**Yongji Zhang**, **Jie Qiu**, **Xianfang Xu and Lingling Zhou** *

Key Laboratory of Yangtze River Water Environment, Ministry of Education, Tongji University, Shanghai 200092, China; yongjizhang@126.com (Y.Z.); stulalala@163.com (J.Q.); xaviera9@163.com (X.X.)
* Correspondence: angelina-zhou@163.com

**Abstract:** With the widespread use of chloramines disinfection, nitrification has become a problem that cannot be ignored. In order to control nitrification in the drinking water distribution system (DWDS), the inactivation effect of free chlorine, monochloramine and chlorine dioxide on ammonia-oxidizing bacterium (AOB) was studied under different temperature (8 °C, 26 °C and 35 °C) and pH (6.0, 7.0 and 8.7) conditions. The inactivation effect of *Nitrosomonas europaea* (a kind of AOB) by the three disinfectants increases with increasing temperature. As for the raised pH, the inactivation effect of free chlorine and monochloramine on AOB decreased, while that of chlorine dioxide increased. Last, but certainly not least, the experimental data of the disinfection were calculated to develop the *N. europaea* inactivation kinetic model, which was based on the first order Chick–Watson equation. The proposed model in this study took the two variables, pH and temperature, into consideration simultaneously, which were used to evaluate the average Ct value (multiplying the concentration of the residual disinfectant by the time of contact with *N. europaea*) required for different disinfectants when they produced the ideal inactivation effect on *N. europaea*.

**Keywords:** ammonia-oxidizing bacterium; *N. europaea*; inactivation; kinetic models; drinking water

## 1. Introduction

The stringent disinfection by-products (DBPs) regulation and potentially carcinogenic identification of DBPs yielded during chlorination in drinking water have instigated the increasingly widespread adoption of chloramines for secondary disinfection [1]. Survey data show that the percentage of facilities using chloramines is predicted to increase to 57% of all surface water and 7% of all groundwater in the United States [2]. In addition, it is estimated that there are more than 30 water utilities throughout China that are adopting chloramination for secondary disinfection in the present period.

Considering the advantages of stronger biofilm penetration, slower decay rate, better dispersion in the DWDS and lower DBPs production than chlorine, chloramines have been regarded as a good alternative for secondary disinfection in drinking water utilities [3–5]. However, nitrification occurring in the DWDS was found in association with chloramination, so it has raised a lot of concerns during the past decades. Nitrification is a kind of microbial reaction; the ammonia released during chloramine decay can trigger nitrification incidence and reduced pH or produce higher nitrite [6,7]. Wilczak et al. mentioned that, among the 67 utilities responding to their telephone surveys, 19% of them experienced moderate nitrification and 4% indicated a severe nitrification problem [8]. Based on Cunliffe's investigation in 1991 [9], AOB was found to exist in 64% of samples collected from five chloraminated water supplies. In addition, the concentration of nitrite was found to reach a seriously high level in the DWDS in one southern city in China [10].

The implementation of chloramination has introduced excess levels of free ammonia to the DWDS, which is associated with chloraminate formation and chloramine decay.

Free ammonia, which serves as an energy source for AOB, is oxidized to nitrite with the growth of these microorganisms and promotes the nitrification process. Nitrification in the DWDS is detrimental and may bring about a series of water environmental issues such as deterioration of water quality, which mainly include the production of nitrite/nitrate, depletion of disinfectant residual, regrowth of heterotrophic bacteria, etc. These problems largely violate the current regulations (e.g., the Surface Water Treatment Rule, the Total Coliform Rule and the maximum contaminant level of 1 mg·L$^{-1}$ N for nitrite). Thus, as the problem of chloramination worsens, the in-depth study on nitrification and its control methods plays an important role in water treatment field.

In order to completely understand DWDS nitrification and its control method, it inherently forces us to explore the origin of this issue, starting with the growth rate of AOB and chloramines disinfection kinetics on AOB.

Unfortunately, only a few studies have addressed the inactivation kinetics on AOB [11–13], which has hampered nitrification research progress. In view of this fact, one of the goals of this study was to measure the disinfection kinetics of free chlorine, monochloramine and chlorine dioxide on AOB at pH values ranging from 6.0 to 9.0 and at a temperature between 8 °C and 35 °C in bench-scale experiments.

In order to further study the nitrification model so that we can better control it, it is necessary to have a thorough understanding of the growth and disinfection kinetics of AOB. Hence, the disinfection kinetic model and mechanism of AOB is an important aspect of this study.

The most common model used in the study of disinfection is the Chick–Watson model [14], which is also the most used model in the study of nitrifying bacteria disinfection models. Oldenburg et al. compared the fitting effects of the Chick–Watson model, Home model and series-event model on the AOB characteristic curves of monochloramine inactivation, and finally found that the Chick–Watson model was the most consistent with the actual data. On this basis, the n-order Chick–Watson model [15] is used to fit the characteristic curve of monochloramine-inactivated *N. europaea* at different pH. All results proved that the first-order Chick–Watson model to be enough to obtain a good fitting effect [11]. Chauret et al. used the deformed first-order Chick–Watson model that fitted the inactivation characteristic curves of free chlorine and monochloramine on *N. europaea* respectively with Ct value and the linear equation of log(N/N$_0$). Results showed that there was a good linear relationship between Ct value and log(N/N$_0$) when the initial disinfectant concentration, the pH value and temperature were constant during the disinfection process [12]. As for the inactivation of *N. europaea* at different concentrations, Wahman et al. used the delayed Chick–Watson model to divide the disinfection situation into two stages. In the first stage, the disinfectant almost did not work when the Ct value ranged from 0 to 500; the value of ln(N/N$_0$) is 0. In the second stage, there is a good linear relationship between Ct value and ln(N/N$_0$) [13].

Considering that the existing AOB inactivation models have to, respectively, evaluate the inactivation rate at a constant pH and temperature combination, this paper presents the AOB inactivation kinetic model based on the Chick–Watson model, in which the inactivation rate constant is expressed as a function of pH and temperature T. It is expected that, as a first step toward a better understanding of AOB inactivation in the DWDS, the modified Chick–Watson disinfection kinetic model including the parameters of pH and temperature can be used to estimate the mean Ct value needed to achieve a desired level of inactivation of AOB in the presence of chlorine or monochloramine and chlorine dioxide, thus improving the practicality of the AOB disinfection kinetic model.

## 2. Materials and Methods

### 2.1. N. europaea and Disinfectants

Previous studies showed that AOB appears to be mixed cultures in a chloraminated DWDS, with the genera Nitrosomonas and Nitrosospira representing the dominant AOB.

This study explored the inactivation kinetics of the pure-culture AOB, establishing the foundation for the future studies on the mixed-culture AOB inactivation kinetics.

*N. europaea* (ATCC19718) was commissioned by China Center of Industrial Culture Collection (CICC) to be purchased from American Type Culture Collection (ATCC). *N. europaea* was incubated in 250 mL glass flasks with 10 mL of broth (ATCC 2265) in the dark at 30 °C for 7 to 10 days. After growth had been established, the cultures were batch grown on a rotary shaker (100 rpm, 30 °C) to stationary phase (7 days) with 10% inoculums (10 mL per 100 mL fresh medium) in 250 mL medium glass flasks.

Free chlorine, monochloramine and chlorine dioxide were applied as disinfectants in this experiment. All disinfectants were prepared before each disinfection experiment by diluting stock solution to a concentration that was equivalent to 1 mg·L$^{-1}$ Cl$_2$. After preparation, the three disinfectants were tested using the *N*,*N*-diethyl-*p*-phenylenediamine (DPD) method; the DPD was purchased form Sigma-Aldrich with a purity of 97%. Residual chlorine and chlorine dioxide were tested using a Pocket Colorimeter residual chlorine detector and chlorine dioxide detector (Hach, Loveland, CO, USA), respectively, and monochloramine was tested using a DR/2800 portable spectrophotometer (Hach, Loveland, CO, USA).

### 2.2. Experimental Protocol for Batch Disinfection Experiments

After inoculation for 7 days, cells were harvested from batch growth by centrifugation at 8000× *g* for 15 min and washed using phosphate buffer saline (PBS) (1×, Corning). This operation was repeated twice followed by resuspension in PBS. For each experiment, washed cells were maintained in a sterile 250 mL chlorine demand-free glass bottle, containing 100 mL PBS buffer to make the bulk solution with the concentration of 10$^5$ CFU·mL$^{-1}$. To avoid cell damage during centrifugation, these washed cell suspensions were then put into a rotating shaker (100 rpm) for 12 h at 30 °C prior to the inactivation experiments. At the beginning of the experiment, free chlorine, monochloramine and chlorine dioxide were added into each flask at the same concentration, equivalent to 1 mg·L$^{-1}$ Cl$_2$, respectively. The mixed solution of Nitrosomonas and chloramine-spiked PBS on a magnetic stir plate was then sampled at selected time intervals. The samples were taken out of the flask and quenched with 100 μL of sterile microtubes (BD Falcon). Each sample was tested for residual chlorine, monochloramine or chlorine dioxide and bacterial viability by flow cytometry.

### 2.3. Flow Cytometry for Viable Bacteria Enumeration

Viable and nonviable *N. europaea* cells were enumerated using the method of flow cytometry (FCM). Comparing with the AOB most probable number (MPN) method requires an incubation period from 21 to 30 days; enumeration using FCM for *N. europaea* cells was more efficient in disinfection experiments. To protect the components from light, SYBR Green I (10,000× stock, s7563, Life Technologies (India) Pvt. Ltd. 306, Agarwal City, Delhi) was diluted with dimethyl sulfoxide (DMSO) (100%, 99.96 atom%D, Sigma-Aldrich, Darmstadt, Germany) to a working stock concentration of 100×. For measurements of intact cell concentrations, 5 μL 100× SYBR Green I and 5 μL propidium iodide (PI, P, Life Technologies Ltd., Auburn, MA, USA) were added into 500 μL samples. After the addition of the dyes, the samples were incubated in the dark for 15 min at 30 °C. After incubation, the samples were analyzed using BD FACSverse with a 488 nm solid state laser. All measurements were performed in duplicate. Green fluorescence was collected in the FITC-A channel at 533 nm and red fluorescence in the 7-AAD-A channel at 670 nm with the trigger set on the green fluorescence. A single, fixed gate was employed as a template together with the corresponding instrument settings. Numbers of *N. europaea* cells contained in this gate after staining with SYBR Green I/PI were the basis for enumeration of viable and nonviable cells. Data were processed using the FlowJo vX.0.7 software, resulting in determinations of the number of viable cells at time t ($N_t$).

## 3. Results and Discussion

### 3.1. Kinetic Analysis of N. europaea Inactivation Experiments

The rate of *N. europaea* inactivation by free chlorine, chloramines or chlorine dioxide was measured using batch experiments at temperatures of 8 °C, 26 °C and 35 °C; the pH values of the system, which refer to a mixed solution of *N. europaea* and spiked PBS solution, were 6.0, 7.0 and 8.5. All the batch experiments were performed at the target disinfectant concentration of 1.0 mg·L$^{-1}$ Cl$_2$ (chlorine disinfection experiments required supplementary free chlorine to maintain target residuals). The number of viable *N. europaea* cells throughout the experiments was assessed using FCM method.

The Chick–Watson model was utilized to describe the results from the batch inactivation experiments in this analysis. The Chick–Watson equation models the $\ln(N_t/N_0)$ as a linear function of Ct and is represented by Equation (1).

$$\ln(N_t/N_0) = -kCt, \tag{1}$$

In which k (>0) is the inactivation rate coefficient (L·mgCl$_2^{-1}$·min$^{-1}$), C is the concentration of residual disinfectant (mg·L$^{-1}$), t is time (min), $N_t$ (CFU·mL$^{-1}$) is the number of viable *N. europaea* cells at time t (CFU·mL$^{-1}$) and $N_0$ is the number of viable *N. europaea* cells at the beginning of the experiments (CFU·mL$^{-1}$). Ct values were calculated by integration of the residual disinfectant concentration (C) up to the given sampling time (t). In order to accurately evaluate Ct values, the concentration fluctuation of residual disinfectant was integrated as a function of time. Then, the Ct is calculated as follow:

$$Ct = \int_0^t Cdt, \tag{2}$$

The Ct values are represented under the curve presented in Figure 1, in which the shaded area was used to estimate the integral term Equation (1).

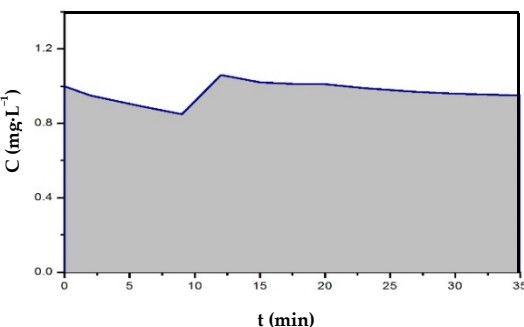

**Figure 1.** Principle schematic diagram of the integration calculation for Ct values.

Best-fit estimates of the k parameters were calculated using the function of Equation (1) at different pH and temperature. In addition, the inactivation efficiencies at different conditions were compared by Ct$_{99}$ values (Ct required for 99% cell inactivation). A summary of Ct$_{99}$ values calculated from the parameters using the Chick–Watson model (Equation (1)) for *N. europaea* inactivation experiments applying free chlorine, monochloramine and chlorine dioxide inactivation at different conditions, which is shown in Table 1, along with the fitted regression coefficient $R^2$ of free chlorine inactivation between 0.94 and 0.99 (Table 1, Figure 2), whereas the $R^2$ is between 0.94 and 0.98 (Table 1, Figure 3) and 0.93–0.98 (Table 1, Figure 4) for monochloramine and chlorine dioxide inactivation, respectively. Among the results above, the inactivation effect and fitting result of monochloramine corresponded well with the existing research results [11,16]. It is suggested that the Chick–Watson model (Equation (1)) is adequate for describing the inactivation of *N. europaea* by these three disinfectants at the pH values and temperature conditions described in this study.

**Table 1.** Effect of pH and temperature on *N. europaea* inactivation.

| Experimental Conditions | | Free Chlorine | | | Monochloramine | | | Chlorine Dioxide | | |
|---|---|---|---|---|---|---|---|---|---|---|
| | | k (L·mg$^{-1}$·min$^{-1}$) | Ct$_{3log}$ (mg·min·L$^{-1}$) | R$^2$ | K (L·mg$^{-1}$·min$^{-1}$) | Ct$_{3log}$ (mg·min·L$^{-1}$) | R$^2$ | K (L·mg$^{-1}$·min$^{-1}$) | Ct$_{3log}$ (mg·min·L$^{-1}$) | R$^2$ |
| pH | 6 | 0.81 | 7 | 0.98 | 0.015 | 463.7 | 0.94 | 0.48 | 14.4 | 0.93 |
| | 7 | 0.35 | 9.2 | 0.97 | 0.009 | 731.7 | 0.98 | 0.83 | 8.3 | 0.96 |
| | 8.7 | 0.22 | 32.9 | 0.99 | 0.006 | 1123.6 | 0.96 | 1.03 | 6.7 | 0.96 |
| Temperature (°C) | 8 | 0.23 | 23.8 | 0.99 | 0.006 | 1132.1 | 0.97 | 0.59 | 11.7 | 0.97 |
| | 26 | 0.35 | 9.2 | 0.97 | 0.009 | 731.7 | 0.98 | 0.83 | 8.3 | 0.96 |
| | 35 | 1.04 | 3.4 | 0.94 | 0.014 | 488.6 | 0.98 | 2.09 | 3.3 | 0.98 |

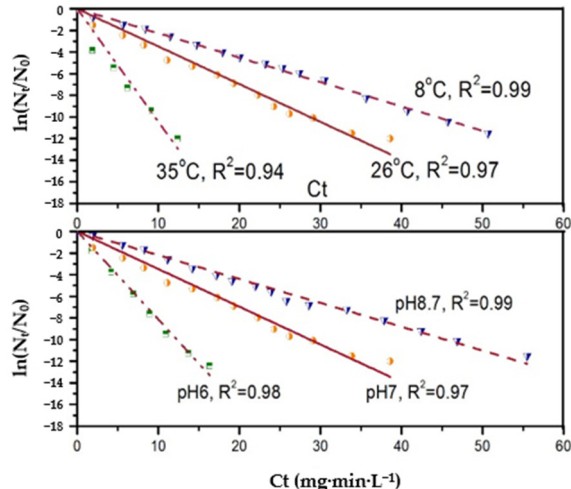

**Figure 2.** Inactivation on *N. europaea* applying free chlorine. Experiments were conducted in PBS buffers at temperature varying from 8 °C to 26 °C to 35 °C and pH varying from 6.0 to 8.7. Linear regressions based on the Chick–Watson model are represented as Equation (1) for different pH and temperature conditions. R$^2$ represents the correlation coefficient values.

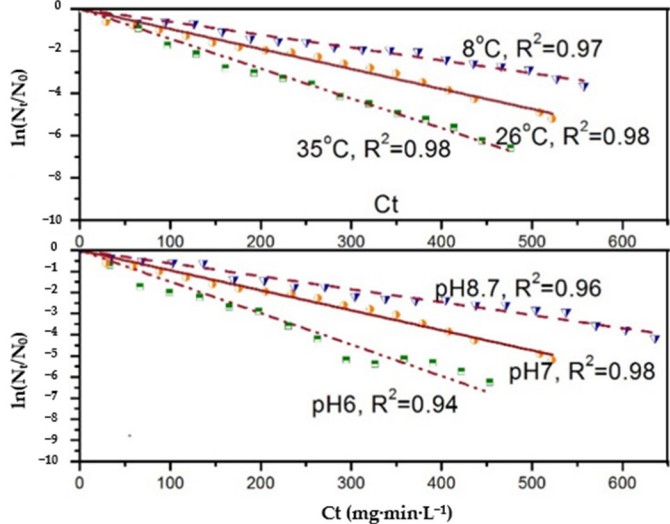

**Figure 3.** Inactivation on *N. europaea* applying monochloramine. Experiments were conducted in PBS buffers at temperature varying from 8 °C to 26 °C to 35 °C and pH varying from 6.0 to 8.7. Linear regressions based on the Chick–Watson model were represented as Equation (1) for different pH and temperature conditions. R$^2$ represents correlation coefficient values.

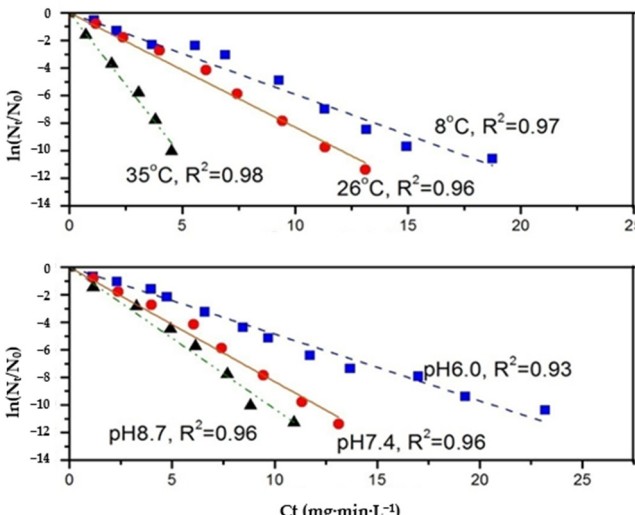

**Figure 4.** Inactivation on *N. europaea* applying chlorine dioxide. Experiments were conducted in PBS buffers at temperature varying from 8 °C, 26 °C to 35 °C, and pH varying from 6.0 to 8.7. Linear regressions based on the Chick–Watson model was represented as Equation (1) for different pH and temperature conditions. $R^2$ represents correlation coefficient values.

### 3.2. Effect of pH and Temperature on N. europaea Disinfection Kinetics

The inactivation effect of pH on *N. europaea* is presented in Table 1 and Figure 5. At the lower pH condition, the results showed that the chlorine or monochloramine inactivation effect on *N. europaea* was more efficient at pH 6.0 and pH 7.0 than that at pH 8.7. However, chlorine dioxide has the strongest inactivation effect at pH 8.7, which is consistent with the trends of chlorine dioxide inactivation of *Pseudomonas aeruginosa* and *Staphylococcus aureus* in water [17]. In other scenarios, the higher k value of *Escherichia coli* inactivation by sequential disinfection with a low level chlorine dioxide followed by free chlorine was obtained at lower pH [18].

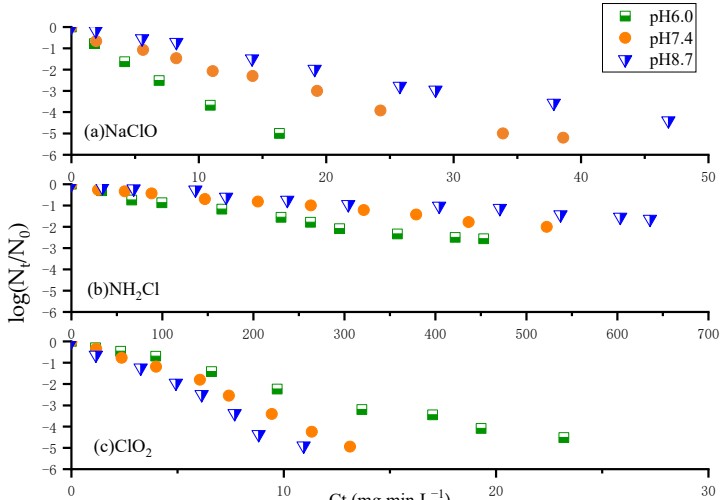

**Figure 5.** Inactivation effect on *N. europaea* using (**a**) free chlorine, (**b**) monochloramine and (**c**) chlorine dioxide at varying pH. Experiments were conducted within the target disinfectant concentration of 1 mg·L$^{-1}$ Cl$_2$. The disinfectant susceptibility of *N. europaea* was analyzed at pH 6.0, pH 7.0 and pH 8.7.

Compared to a Ct value of 32.9 mg·min·L$^{-1}$ at pH 8.7, a Ct$_{99}$ value of 7.0 mg·min·L$^{-1}$ was acquired at pH 6.0 and 9.2 mg·min·L$^{-1}$ at pH 7.0 using free chlorine; compared to a Ct value of 1123.6 mg·min·L$^{-1}$ at pH 8.7, a Ct$_{99}$ value of 463.7 mg·min·L$^{-1}$ was yielded at

pH 6.0 and 731.7 mg·min·L$^{-1}$ at pH 7.0 using monochloramine. The Ct value of chlorine dioxide disinfection was 6.7, 8.3 and 14.4 mg·min·L$^{-1}$ at pH 8.7, pH 7.0 and pH 6.0, respectively.

The k-values of free chlorine and monochloramine disinfection at pH 6.0 were significantly higher than those at pH 8.7, reaching 3.7 times (0.81 L·mg$^{-1}$·min$^{-1}$) and 2.5 times (0.015 L·mg$^{-1}$·min$^{-1}$), respectively. However, the k-value of chlorine dioxide disinfection at pH 8.7 (1.03 L·mg$^{-1}$·min$^{-1}$) is more than two times higher than that at pH 6.0 (0.48 L·mg$^{-1}$·min$^{-1}$). As previous research reveals [19], HOCl produced during the chlorination or chloramination plays a main role in disinfection. More HOCl will be present at a lower pH level, which leads to the observed increase in inactivation efficiency. This is consistent with the fact that HOCl accounts for an increasingly higher proportion of the total chlorine concentration as pH reduces. Chlorine dioxide inactivates bacteria by the action of single molecules, which means its mechanism is different from that of free chlorine and monochloramine. A higher pH level can increase the concentration of hydroxide ions, thus playing a catalytic role in the disinfection effect of chlorine dioxide [20]. Therefore, the effect of the change in pH on the inactivation of AOB by chlorine dioxide showed an opposite trend to that of free chlorine and monochloramine.

The inactivation rate for *N. europaea* varied according to the temperature variation as demonstrated in Table 1 and Figure 6. Based on the chart analysis, an increase in temperature (from 8 °C to 35 °C) resulted in better inactivation (Figure 6). The Ct$_{99}$ value obtained with free chlorine increased from 3.4 mg·min·L$^{-1}$ at 35 °C to 23.8 mg·min·L$^{-1}$ at 8 °C; whereas, for monochloramination, the Ct$_{99}$ value increased from 488.6 mg·min·L$^{-1}$ at 35 °C to 1132.1 mg·min·L$^{-1}$ at 8 °C. In addition, the Ct$_{99}$ value of chlorine dioxide disinfection increased from 3.3 mg·min·L$^{-1}$ at 35 °C to 11.7 mg·min·L$^{-1}$ at 8 °C. Inactivation of suspended AOB by these three disinfectants increased with increasing temperature. As can be seen from the trend of the changes in curves in Figure 6, the effect of temperature on free chlorine and chlorine dioxide was more significant than that of monochloramine. As mentioned before, HOCl, the hydrolysis products of disinfectants, played a main role in the inactivation process. Plus, it also improved with increasing temperature [19].

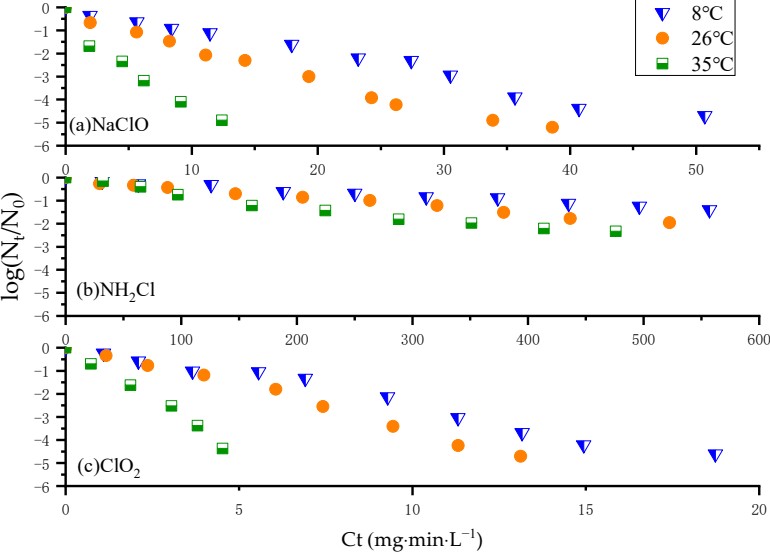

**Figure 6.** Inactivation effect on *N. europaea* using (**a**) free chlorine, (**b**) monochloramine and (**c**) chlorine dioxide at varying temperatures. Experiments were conducted within the target disinfectant concentration of 1 mg·L$^{-1}$ Cl$_2$. The disinfectant susceptibility of *N. europaea* was analyzed at 8 °C, 26 °C and 35 °C.

### 3.3. Development of the Disinfection Kinetic Model

The Chick–Watson model has been proven to be effective in many studies; its mathematical properties and application to disinfectants, such as chlorine, ozone, hydrogen peroxide, copper and ethanol, have been described and discussed [21–25]. However, it requires separately determining the disinfection rate constant for each pH and temperature combination [14,15,26], which may reduce the versatility of the model. Therefore, this equation model was modified when inactivation rate constant k was integrated as a function of pH and temperature, yielding the following functional relationship:

$$K = R(\alpha)^{PH}(\beta)^{T} \tag{3}$$

in which R, $\alpha$ and $\beta$ are constants to be determined by regression analysis. Input Equation (3) into Equation (1) to obtain Equation (4):

$$-\ln(N_t/N_0) = R(\alpha)^{PH}(\beta)^{T}(Ct) \tag{4}$$

Taking the ln of both sides of Equation (3):

$$\ln[-\ln(N_t/N_0)] = \ln(R) + pH\ln(\alpha) + T\ln(\beta) + \ln(Ct) \tag{5}$$

Setting:

$$Y = \ln[-\ln(N_t/N_0)/Ct] \tag{6}$$

Then yields:

$$Y = \ln(R) + pH\ln(\alpha) + T\ln(\beta) \tag{7}$$

In this study, the authors used the fitting function of MATLAB software to fit the *N. europaea* disinfection kinetic models including the parameters of pH and temperature, proposing the following models:

kinetics of free chlorine disinfection:

$$\ln(N_t/N_0) = -7.14(0.59)^{PH}(1.04)^{T}(Ct) \tag{8}$$

kinetics of monochloramine:

$$\ln(N_t/N_0) = -0.07(0.7)^{PH}(1.03)^{T}(Ct) \tag{9}$$

kinetics of chlorine dioxide:

$$\ln(N_t/N_0) = -0.05(1.32)^{PH}(1.03)^{T}(Ct) \tag{10}$$

in which the variables are as defined previously.

### 3.4. Model Validation

Figure 7 is a plot of the comparison of the predicated values and the measured data of three disinfectant models. As can be seen, the measured data are close to the model predictions.

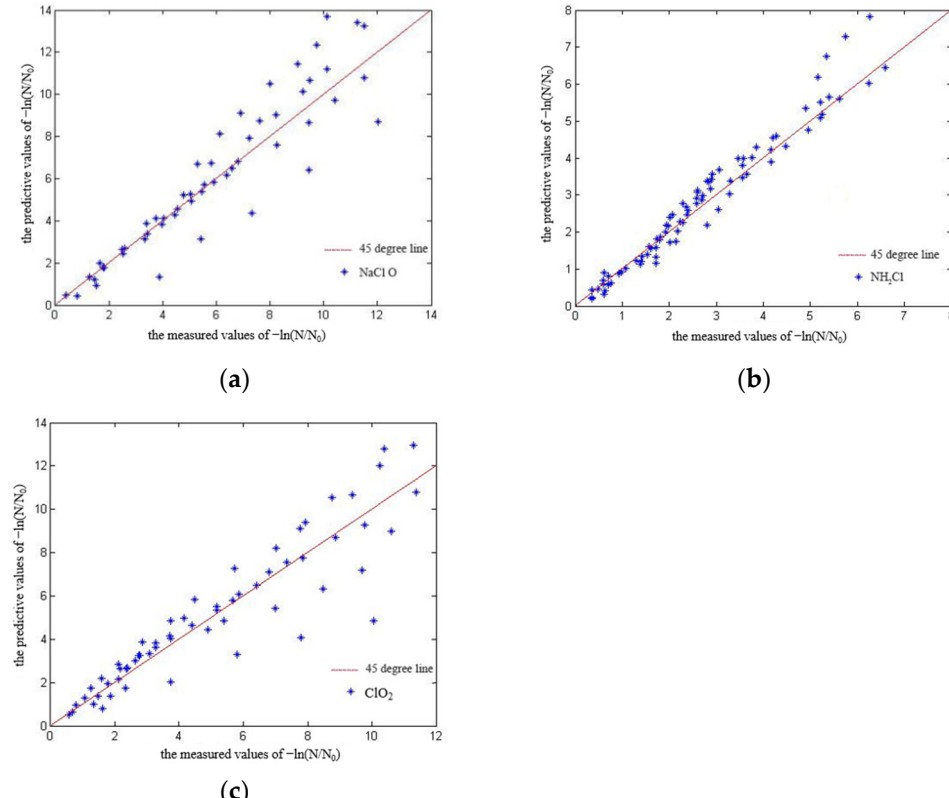

**Figure 7.** Predictive and measured values of (**a**) NaClO, (**b**) NH$_2$CL or (**c**) ClO$_2$ disinfection model.

The prediction performance of the models was expressed with correlation coefficient and Root Mean Square Error (RMSE). Table 2 illustrates the correlation coefficient (r) and RMSE of the three disinfection models.

**Table 2.** Correlation coefficient and RMSE of two disinfection models.

| Disinfectant | r | RMSE |
|---|---|---|
| Free chlorine | 0.929 | 0.234 |
| monochloramine | 0.978 | 0.046 |
| Chlorine dioxide | 0.917 | 0.168 |

According to Figure 7 and Table 2, the correlation coefficient of the disinfection kinetic model was 0.929 for free chlorination, 0.978 for monochloramination and 0.917 for chlorine dioxide, indicating a positive correlation between the predictive and measured values of the two disinfectant models. The RMSE of free chlorine, monochloramine disinfection and chlorine dioxide model was 0.234, 0.046 and 0.168, respectively, suggesting a low dispersion degree of the predictive and measured values of the disinfectant models. In conclusion, the disinfection kinetic model including the parameters of pH and temperature developed in this study can readily predict the disinfection effects on *N. europaea* by free chlorine, monochloramine or chlorine dioxide.

## 4. Conclusions

(1) The inactivation effect of *N. europaea* by the three disinfectants increases with increasing temperature. Free chlorine and chlorine dioxide react more significantly to increased temperature than monochloramine. The inactivation effect of these three disinfectants on *N. europaea* showed different trends with increasing pH. Among them, the inactivation effect of free chlorine and monochloramine on *N. europaea* decreased, while that of chlorine dioxide increased.

(2)　The inactivation effects of the three disinfectants on *N. europaea* were ranked in descending order as chlorine dioxide, free chlorine and monochloramine, with monochloramine being much less effective than the others. There is a potential risk of nitrification in the DWDS when monochloramine is used for disinfection; monochloramine can stop being added regularly and strong disinfectants such as free chlorine and chlorine dioxide can be used to inactivate AOB.

(3)　The inactivation coefficient k-value of disinfectant on *N. europaea* varied with pH and temperature, and the effect of pH and temperature on the k-value was not linear.

(4)　The inactivation coefficient k-value was expressed as a function of pH and temperature, and it was substituted into the Chick–Watson model to fit the disinfection prediction model containing pH and temperature T. This model can better quantify the inactivation process of AOB in three disinfectants from the perspective of kinetics.

**Author Contributions:** Conceptualization, Y.Z. and J.Q.; methodology, Y.Z. and J.Q.; software, X.X.; validation, Y.Z., J.Q. and X.X.; formal analysis, Y.Z., J.Q. and X.X.; investigation, X.X. and J.Q.; resources, Y.Z., J.Q. and X.X.; data curation, X.X. and J.Q.; writing—original draft preparation, J.Q.; writing—review and editing, Y.Z., J.Q., X.X. and L.Z.; visualization, X.X.; supervision, Y.Z.; project administration, Y.Z. and L.Z.; funding acquisition, Y.Z. and L.Z. All authors have read and agreed to the published version of the manuscript.

**Funding:** This work was funded by the National Natural Science Foundation of China (Grant Nos. 51778453 and 51878467).

**Informed Consent Statement:** Not applicable.

**Data Availability Statement:** Not applicable.

**Acknowledgments:** This work was supported by the National Natural Science Foundation of China (Grant Nos. 51778453 and 51878467).

**Conflicts of Interest:** The authors declare no conflict of interest.

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
