# Peer review of "Disinfection Kinetics of Free Chlorine, Monochloramines and Chlorine Dioxide on Ammonia-Oxidizing Bacterium Inactivation in Drinking Water"

_water, doi:10.3390/w13213026_

Round 1

Reviewer 1 Report

The manuscript water-1403438 examined the inactivation effect of three disinfectants (free chlorine, monochloramine and chlorine dioxide) on ammonia oxidizing bacterium N. europaea under different temperature (8℃, 26℃ and 35℃) and pH (6.0, 7.0 and 8.7) conditions. Moreover, an inactivation kinetic model, taking into consideration the pH and temperature values, was developed and found to be useful in describing the observed kinetics of several disinfectants. Their findings are helpful to better understand and control the DWDS nitrification process. However, there are some overall suggestions for the authors. I suggest acceptance with minor revision.

  1. How many days lasted the inactivation experiments in PBS solution?
  1. Line 152: The selected pH values refer to mixed solution of N. europaea and spiked PBS solution? Please clarify.
  1. Lines 155-156: "The number of viable N. europaea cells throughout the experiments was assessed using FCM method". Thus, the non-viable cells was the difference of viable from the initial concentration?
  1. Line 162: Add "Cl2" after "residual".
  1. Line 182: Add "is" before "suggested".
  1. Lines 217-224: Although, disinfection efficiencies were expected to follow the trend: ClO2>Cl2>NH2Cl, at the lowest pH value (pH 6.0) the opposite was observed. Authors clearly state possible reasons for the different effects of pH changes on AOB inactivation by the selected disinfectants. However, I suggest adding relevant reference/s to support those statements on disinfection mechanisms.
  1. Lines 237-238: "The effect of temperature on free chlorine and chlorine dioxide was more significant than that of chloramine." Any reason why? Please explain.

Author Response

Thanks for your suggestions, here are my response to your comments:

1. As to inactivation duration, all disinfectants were prepared before each disinfection experiment by diluting stock solution to a concentration which equivalent to 1 mg·L-1 Cl2.  And the abscissa of the figures are displayed as Ct values, so the inactivation duration can be calculated, thanks.

2. The selected pH values did refer to mixed solution of N. europaea and spiked PBS solution. I have clarified it in the article.

3. Yes, it is. The FCM method was used to count the number of cells quickly and efficiently, and compared that to the the initial concentration.

4. Revised.

5. Revised.

6. Thanks for your suggestions, however, I tried to find some relevant reference but failed.

7. Explanation has been added in the article. ' As can be seen from the trend of the changes of curves in the Figure 6, the effect of temperature on free chlorine and chlorine dioxide was more significant than that of chloramine. '

Reviewer 2 Report

Manuscript ID: water-1403438

Disinfection kinetics of free chlorine, monochloramines and chlorine dioxide on ammonia-oxidizing bacterium inactivation in drinking water

Yongji Zhang, Jie Qiu, Xianfang Xu, Lingling Zhou *

Submitted to section: Water Use and Scarcity, Advanced Technologies in Drinking Water Treatment, Algae and Disinfection By-Products Control

The article presented for review concerns a very important research topic related to the security of society. Drinking water quality has a wide impact on the health of the population. Water supply systems are a blessing for mankind, but it is necessary to constantly control the water produced in them. Microbiological purity is closely related to the chemical purity of water, and a number of disinfectants are used in this regard. In the presented study, the effectiveness of three disinfectants against ammonia-oxidizing bacteria was investigated. Work is generally well written and documented. However, it requires some improvements before it can be accepted for publication.

  1. Quotations are missing from passages presenting relevant information: lines 42, 56, 61, 66, 72, 75.
  2. Information in heading 2 is incomplete. There is a lack of data on the purity of the reagents and where some reagents and materials were purchased (e.g. phosphate buffer, DPD, DMSO, shaker, falcons).
  3. More detailed information is needed on the bacterial strains used, mainly where they were obtained.
  4. The heading row in table 1 should be corrected. The words in the headings should start with a capital letter, symbols should be written in italics, etc. Please correct the notation of powers and units!
  5. What is Ct? The abbreviation is used from the abstract and no explanation.
  6. In table and figure captions or footnotes, explain any abbreviations and symbols used in the object.
  7. Figures 5 and 6 - too large markers were used to mark the results of the experiment and thus the results are not readable.

Author Response

Thanks for your suggestions, here are my response:

  1. Quotations have been added.
  2. The manufacturer and purity of the reagents have been completed.
  3.  ' N.europaea (ATCC19718) was commissioned by China Center of Industrial Culture Collection (CICC) to be purchased from American Type Culture Collection (ATCC). ' has been added in the article.
  4. Table 1 has been revised, I hope the editor could help me optimize it to be better.
  5. The explanation of Ct value has been given in the abstract.
  6. Revised.
  7. Markers in Figures 5 and 6 have been resized to make the results more readable.

Reviewer 3 Report

Authors describe disinfection kinetics of free chlorine. However, the R&D section lacks comparison to the existing literature on the disinfection processes. There is a lot of literature on the disinfection processes and their mechanisms. Authors need to rewrite the R&D section.

Author Response

Thanks for your time and suggestions, some revisions and additions have been make to the R&D section, including some extra references. However, there are not many studies focusing on the inactivation of ammonia-oxidizing bacterium. So, if you have any comments or suggestions, please let me know. Thank you very much again.

Round 2

Reviewer 3 Report

The authors have added a few references to the R&D section. However, section 3.3 still needs some comparison to the literature. Chick and Watson's model is well studied and reported.

Author Response

Thanks for your time and suggestions again. Several comparisons and literatures concerned about dynamic kinetics of Chick-Watson model have been added to the section 3.3.